# Game Theoretic Honeypot Deployment in Smart Grid

**DOI:** 10.3390/s20154199

**Published:** 2020-07-28

**Authors:** Panagiotis Diamantoulakis, Christos Dalamagkas, Panagiotis Radoglou-Grammatikis, Panagiotis Sarigiannidis, George Karagiannidis

**Affiliations:** 1Department of Electrical and Computer Engineering, Aristotle University of Thessaloniki, 54124 Thessaloniki, Greece; padiaman@auth.gr (P.D.); geokarag@auth.gr (G.K.); 2Testing, Research and Standards Centre, Public Power Corporation S.A., 15351 Athens, Greece; c.dalamagkas@dei.com.gr; 3Department of Electrical and Computer Engineering, University of Western Macedonia, 50100 Kozani, Greece; pradoglou@uowm.gr

**Keywords:** smart grid, cybersecurity, honeypots, game theory

## Abstract

The smart grid provides advanced functionalities, including real-time monitoring, dynamic energy management, advanced pricing mechanisms, and self-healing, by enabling the two-way flow of power and data, as well as the use of Internet of Things (IoT) technologies and devices. However, converting the traditional power grids to smart grids poses severe security challenges and makes their components and services prone to cyber attacks. To this end, advanced techniques are required to mitigate the impact of the potential attacks. In this paper, we investigate the use of honeypots, which are considered to mimic the common services of the smart grid and are able to detect unauthorized accesses, collect evidence, and help hide the real devices. More specifically, the interaction of an attacker and a defender is considered, who both optimize the number of attacks and the defending system configuration, i.e., the number of real devices and honeypots, respectively, with the aim to maximize their individual payoffs. To solve this problem, game theoretic tools are used, considering an one-shot game and a repeated game with uncertainty about the payoff of the attacker, where the Nash Equilibrium (NE) and the Bayesian NE are derived, respectively. Finally, simulation results are provided, which illustrate the effectiveness of the proposed framework.

## 1. Introduction

The recent adoption of innovative Internet of Things (IoT) technologies and products led to the evolution of several domains of critical infrastructures, including health, transportation, and utilities. Power grids, in particular, have been enhanced with Information and Communication Technologies (ICT) at operational and resiliency level with new smart functionalities, including real-time monitoring, smart management, smart customer billing, and provisioning of resources to normalize fluctuations and address unexpected events. Smart meters, phasor measurement units, smart relays, remote terminal units (RTUs), and Programmable Logic Controllers (PLCs) are only a few of the IoT devices that are utilized by energy operators in order to convert traditional power grids to smart grids.

However, the introduction of all these new IoT devices has side effects, including an increasing attack surface. According to the Cisco Annual Internet Report for 2018–2023 [1], it is estimated that Distributed Denial of Service (DDoS) attacks will double to 15.4 million by 2023, which is expressed in 14% Compound Annual Growth Rate (CAGR). The statistics coming from the energy sector are also worrisome. According to LNS Research, 53% of industrial stakeholders have reported experiencing a cyberattack in the last 12 months [2] and “76% of energy executives cited business interruption as the most impactful cyber loss scenario for their organization” [3]. It is evident that even though the research on cybersecurity is progressing rapidly and the market stakeholders pursue the adoption of new cybersecurity products, cyber threats have an increasing trend.

The research community has provided innovative solutions to tackle cyber threats in the critical infrastructure and the energy domain, including intrusion detection systems and threat information sharing platforms that leverage Artificial Intelligence (AI) and modern cryptography techniques. The H2020-DS-SC7-2017 SPEAR: Secure and PrivatE smArt gRid project is a research project, funded by the European Commission, intends to provide a complete cybersecurity solution for modern smart grids by integrating AI-enabled anomaly detection, visual analytics, reputation schemes, forensic investigation frameworks, and deception mechanisms [4].

Even though security mechanisms like signature-based and behavioral-based anomaly detection dominate in the cybersecurity domain, honeypots are emerging as an alternative strategy to trap intelligent cyberattackers that bypass traditional security measures. A first widely accepted definition of honeypots is provided by Spitzner [5]: “A honeypot is a decoy computer resource whose value lies in being probed, attacked, or compromised.” Honeypots are deployed by organizations to disorient cyberattackers that target the infrastructure in production and persuade them to attack the honeypots rather than the real infrastructure. This can serve multiple purposes: either to prevent attacks against the valuable assets or to collect intelligence about the attacker’s activity. These deployment options are known as production and research honeypots, respectively [6].

A major drawback of honeypots is that, during their operation, they reserve resources in a constant manner, regardless of the attacker’s activity, if any. Therefore, a large number of honeypots may lead to resource wastage, while a small number of honeypots may result to inefficient defenses to potential cyberattackers, thus resources are also wasted in this case as the invested resources do not accomplish their purpose. This practical quandary that security engineers face with honeypot orchestration is an active research issue, and game theory has been proposed to enable dynamic configuration of honeypots, by providing the optimal strategy for the defender, taking into account that the adversary is rational and tries to maximize his payoff. Our research aims to address the issue of honeypot orchestration by focusing on smart grid systems and considering their unique characteristics [7].

### 1.1. Related Works & Motivation

Game theory and its potential applications have been thoroughly studied in the context of cyber security [8] and honeypot deployment [9,10], although there is a lack of realistic schemes. The following paragraphs provide an overview of frameworks related to honeypot deployment and orchestration that were considered for our work.

Denial of Service (DoS) and Distributed DoS (DDoS) attack scenarios gain significant attention in the literature as their detection and mitigation is still an open research issue in the domain of cybersecurity. Thus, many of the existing game theory models focus on such type of attacks and provide specific strategies to confront such threats [11]. In more detail, Ceker et al. [12] have proposed a deception-based defense framework to tackle DoS attacks as well as threats that may employ unconventional stealth methods. The proposed framework provides a game-theoretical approach to model the interaction between the defender and the attacker, while a proactive deception mechanism is employed in this dynamic game to confuse the attacker about the defender’s profile. The deception mechanism is based on a Bayesian signaling game of incomplete information, and the perfect Bayesian Equilibrium is utilized as a solution of the proposed framework that takes into consideration resource constraints. The analytical results study the relation between invested resources, processing cost, and the desired security level. Even though the proposed game provides a dynamic framework that scales to other kinds of attacks, except DoS, it is highlighted by the authors that several limitations apply, one of them is that legitimate users may be blocked by the defender, while the defender cost is constant and could be converted to a dynamic function that reflects the implications of the decided actions in a more realistic way.

Wang et al. [13] investigated the deployment of honeypots in an Advanced Metering Infrastructure (AMI), a typical network architecture utilized by Distribution System Operators (DSO) to obtain measurements from smart meters in the modern smart grid. The proposed game aims to address DDoS attacks in the aforementioned network topology, and to this aim, they introduce a Bayesian game model to find the equilibrium between legitimate users and attackers. An AMI network with four service providers, 10 honeypots, and two anti-honeypots is simulated via OPNET to obtain evaluation results, which indicate the optimal number of honeypots to be deployed with the given parameters when a balance between detection rate and energy consumption is achieved. It is also highlighted by the authors that the effectiveness of the defense strategy does not necessarily improve when more honeypots are deployed.

The promising and innovative concept of Software-Defined Networks (SDN) is adopted in [14] to propose a game-theoretic framework that estimates the optimal strategies for both defenders and attackers, considering the balance between energy consumption and detection rate. As highlighted by the authors, the centralized nature of SDN makes the architecture susceptible to (D)DoS attacks, and the proposed model aims to deploy a defense mechanism against such attacks. Moreover, anti-honeypot attacks and pseudo-honeypot game strategies are introduced in this research to model and tackle DDoS attacks, respectively, resulting in several Bayesian Nash solutions. To evaluate the proposed model, a realistic testbed was constructed with hosts, attackers, and OpenFlow switches. The experimental results outperform in terms of performance in energy consumption and detection rate.

Al-Shaer et al. [15] proposed a different approach, compared to the previously mentioned references, that is based on the hypergame theory. The main motives of this perspective are the capabilities that are offered in terms of defense strategies for both proactive and reactive approaches as well as the limited contribution in the literature regarding mature and well-structured mathematical frameworks listing the hypergame concept. In the proposed work, an attack–defense model is structured with subjective beliefs in a dynamic environment, in which the defender tries to manipulate the attacker’s belief utilizing deception techniques. Hypergame theory provides the ability to estimate the decision of each player and the impact that the uncertainty has on the expected utility. The deception model is studied by modeling a Stochastic Petri Net and the results deliver insightful findings that relate the perceptions by different players (i.e., an attacker or a defender) with their chosen optimal strategies and the corresponding utilities.

A Partially Observable Stochastic Game (POSG) was introduced in [16] that applies in situations where each player has partial information about the environment. In particular, the authors develop a POSG-based game theoretic framework to optimize honeypot deployment that assumes literal movement of the attacker in a computer network. The attacker and the defender are placed on a graph, in which nodes represent network hosts and the edges represent attacks against other hosts, with each attack incurring an associated cost. In this context, the attacker tries iteratively to attack hosts, while the defender chooses the edges that will act as honeypots. The experimental results prove that the POSG model was able to generate near-optimal deployment strategies as well as realistic and scalable networks of multiple hosts.

The authors of [17,18] proposed a game-theoretic framework that focuses on Cyber-Physical system (CPS) honeypots, with both low and high interaction. The proposed model is specifically used to deploy defensive mechanisms against Advanced Persistent Threats (APTs) in CPS and considers limited resources for honeypot allocation and human analysis as well as incomplete information for the players. Simulation results prove that the proposed model succeeds to maximize the defender’s payoff and provides multiple Bayesian equilibria.

The authors of [19,20] used game theory to study various attacks and defense scenarios in networks with honeypots. Specifically, they utilize a Bayesian model to adequately reflect the defender’s imperfect knowledge of user behavior (i.e., normal or malicious), thus forming a Bayesian signaling game of incomplete information. A one-shot game model is presented in order to determine how the defender should react to different user behaviors. Moreover, the authors provided a repeated version of this game that enables the players to update their opinions under a Bayes rule. Finally, mathematical analysis, as well as simulations, are used to find the equilibria and further evaluate the model. The results suggest that when the defender is facing attacks with high frequency, the best action is to massively deploy honeypots. Otherwise, in the case of low-frequency attacks, the defender can mix up their strategy.

Finally, Bilinski et al. in [21] investigated the Nash Equilibrium (NE) of a honeynet system, in which the defender aims to protect a number of network hosts and has a fixed set of resources, therefore can defend only a limited number of hosts. On the contrary, the attacker can attack a specific number of hosts concurrently, although no cost is incurred to the adversary for each attack. In this context, the attacker is considered the winner if they attack a real host and not a honeypot, otherwise the defender wins. The analysis of the proposed model concludes that the value of a host is inversely proportional to the probability of the host to being attacked. However, certain limitations are remarked, including the fact that the attacker’s activity is not limited by a cost function and that the proposed game assumes that the attacker wins the game if it attacks any host which is not a honeypot as well as that the number of served real devices is fixed, despite the fact that limited resources have been assumed. Finally, a similar scenario has been investigated in [22], assuming though that the defender has complete information about the attacker’s payoff, a cost function for the attacker, and fixed number of served real devices, which has led to the formulation of a Stackelberg game. Moreover, in contrast in [21], the payoff of the attacker has been considered to be an increasing function of the number of attacked real devices.

### 1.2. Contribution

In this paper, game theory is used to model the interaction between an attacker and a defender, who makes use of honeypots to mitigate the impact of attacks within a smart grid. Taking into account the trade-off between connectivity and security, which is an important challenge in the smart grid, a novel framework is proposed according to which the defender has the option to periodically substitute part of the real devices with honeypots, e.g., for a portion of time, with the aim to deceive the attacker. More specifically, the defender optimizes the number of connected real devices and honeypots, taking into account the attacker’s preferences. First, we focus on one encounter between the attacker and the defender, which is solved by using the concept of NE. Moreover, an alternative optimization framework is proposed for the case that the NE does not exist. Next, we extend the analysis considering a more sophisticated attacker, who randomizes its strategy, by attacking a random number of hosts, while also considering a repeated game and uncertainty about the attacker’s payoff parameters. In this case, the interaction between the attacker and the defender is modeled as a multi-stage Bayesian game and the Bayesian NE is derived. Moreover, a rule to update the defender’s belief about the type of the attacker is also provided. Finally, simulation results are provided to illustrate the effectiveness of the proposed framework.

### 1.3. Structure

The rest of the paper is organized as follows. The system model for the strategies and payoffs of the attacker and the defender is introduced in Section 2. In Section 3, the NE in the case of a one-shot game is derived, while the case that the NE does not exist is also discussed. In Section 4, the results of Section 3 are extended to the case of repeated games with uncertainty about the type of the attacker. Simulation results are given and discussed in Section 5. Finally, conclusions are summarized in Section 6.

## 2. System Model

A defending system is considered within a smart grid, hereinafter termed as a defender, that protects a collection of hosts from a potential attacker by using honeypots, which are able to detect unauthorized accesses, collect evidence, and help hide the real devices [13]. The honeypots are designed to mimic common services of the smart grid, including Industrial Control System (ICS) devices, smart meters, and smart appliances, among others [7]. Figure 1 depicts the most common locations of attack threats and honeypots as well as real-life applications of the proposed model. In more detail, honeypots could be deployed in Supervisory Control and Data Acquisition (SCADA) networks, located in smart factories, power plants, and Distributed Energy Resources (DERs), to mimic various ICS devices, including Programmable Logic Controllers (PLCs), sensors, and smart relays, among others [23]. Moreover, honeypots could be deployed in substations owned by Transmission System Operators (TSO) or Distribution System Operators (DSO) to emulate more advanced ICS devices, including Remote Terminal Units (RTUs) and Phasor Measurement Units (PMUs). Finally, honeypots could be applicable in smart buildings to emulate smart appliances and energy meters or in Advanced Metering Infrastructures (AMIs), operated by DSOs, to emulate smart meters [24]. It is assumed that the defender has a fixed set of resources and, as a result, is only able to defend a limited number of hosts [21].

Considering the proposed system model, the corresponding attack model ensembles a wide range of attack scenarios, especially those that target specific vulnerable network assets and in which the adversary has to choose between assets in the operational environment and honeypots. In more detail, DoS attacks are very common in smart grid applications and include a variety of attacks, such as buffer overflow, flooding, and amplification attacks, among others, that aim to render a remote service inaccessible to legitimate users. By its definition, the proposed system model aggregates possible multiple adversaries to a single entity, therefore the attack model also considers DDoS attacks, where multiple systems launch orchestrated attacks against a single host. Finally, False Data Injection Attacks (FDIAs) can also be considered for the attack model as they target specific assets in a smart grid. FDIAs aim to tamper control systems with falsified data that can manipulate the decision of automation systems, with severe consequences ranging from the destruction of smart grid equipment to grid fluctuations, instabilities, and financial loses [25].

Let $N\leq N_{\max}$ denote the total number of hosts within a block of IP addresses, with the value of *N* being controlled by the defender. Additionally to the total number of hosts, the defender can also control which of them are used by real devices and honeypots, with the aim to mitigate the impact of potential attacks without unnecessarily increasing the related costs. It is highlighted that in the considered scenario, the defender has the option to increase the number of honeypots by disconnecting real devices (each of which for a portion of time), if this further assists on further mitigating the impact of potential attacks. This approach aims at exploring the potential security gains of “hiding” some of the real devices and substituting them with honeypots. In general, the defender’s decision is affected by several parameters, such as the deployment costs; the benefit of capturing an attack with a honeypot; the cost of having a number of real devices under attack; and the trade-off between increasing the number of real devices that are connected to the smart grid at each time slot, the level of security, which increases with the number of utilized honeypots, and the implementation cost.

The attacker set of strategies determines whether or not to attack a host. Thus, the attacker’s decision depends on the trade-off between the benefit acquired when attacking a real device and the cost of attacking a honeypot.

For the *t*-th interval, let $s_{D,i}\left[t\right]\in \left\{1,-1\right\}$ be equal to 1 when the *i*-th host is used by a real device and equal to −1 when it is used by a honeypot. On the other hand, regarding the set of strategies of the attacker, let $s_{A,i}\left[t\right]\in \left\{1,0\right\}$ be equal to 1 when the attacker attacks the *j*-th host and equal to 0 when the *j*-th host is not attacked. For the sake of clarity, the notation is given in Table 1.

According to the aforementioned trade-off for the attacker’s side, the attacker’s payoff is given by
(1)$$U_{A}\left[t\right]=f\left(a_{i\in \{1,2,3\}},\sum_{i=1}^{N}\frac{\left(1+s_{D,i}\right)}{2}s_{a,i},\sum_{i=1}^{N}\frac{1-s_{D,i}}{2}s_{a,i},\sum_{i=1}^{N}s_{A,i}\right),$$
where $a_{1},a_{2},a_{3}$ are the non-negative weights that correspond to the impact that the number of attacked real devices; the number of attacked honeypots and the total number of attacks has on its payoff; and *f* is an increasing function of $\sum_{i=1}^{N}\frac{\left(1+s_{D,i}\right)}{2}s_{A,i}$, i.e., the number of attacked real devices, and a decreasing function of $\sum_{i=1}^{N}\frac{1-s_{D,i}}{2}s_{A,i}$, i.e., the number of attacked honeypots. Moreover, the total number of attacks, i.e., $\sum_{i=1}^{N}s_{A,i}$, also introduces extra cost to the attacker’s payoff due to the implementation cost and the general increase of the probability of the attacker to reveal information about their identity and action. For example, assuming that the aforementioned terms have a linear impact on the attacker’s payoff, $U_{A}\left[t\right]$ could be written as
(2)$$U_{A}\left[t\right]=a_{1}\sum_{i=1}^{N}\frac{\left(1+s_{D,i}\right)}{2}s_{A,i}-a_{2}\sum_{i=1}^{N}\frac{1-s_{D,i}}{2}s_{A,i}-a_{3}\sum_{i=1}^{N}s_{A,i}.$$

On the other hand, the defender’s payoff is given by
(3)$$U_{D}\left[t\right]=g\left(d_{i\in \{1,2,3,4\}},\sum_{i=1}^{N}\frac{\left(1-s_{D,i}\right)}{2}s_{A,i},\sum_{i=1}^{N}\frac{\left(1+s_{D,i}\right)}{2}s_{A,i},\sum_{i=1}^{N}\frac{\left(1+s_{D,i}\right)}{2},N\;\;\right),$$
where $d_{1},d_{2},d_{3},d_{4}$ are the non-negative weights that correspond to the impact that the number of attacked real devices, the number of attacked honeypots, the number of real devices that are not served, and the total number of used hosts has on its payoff, and *g* is an increasing function of $\sum_{i=1}^{N}\frac{\left(1-s_{D,i}\right)}{2}s_{A,i}$ and a decreasing function of the absolute value of $\sum_{i=1}^{N}\frac{\left(1+s_{D,i}\right)}{2}-N_{r}$. Moreover, the total number of hosts also introduces an extra cost. Next, it is assumed that the terms coupled with $d_{1}$, $d_{2}$, and $d_{4}$ have a linear impact on the attacker’s payoff. Moreover, it is considered that the level of satisfaction of the defender gradually gets saturated as more real devices are served, i.e., the defender’s payoff is a concave function, hereinafter modeled by square function, of the number of real devices that are not served. Thus, $U_{D}\left[t\right]$ could be written as
(4)$$U_{D}\left[t\right]=d_{1}\sum_{i=1}^{N}\frac{\left(1-s_{D,i}\right)}{2}s_{A,i}-d_{2}\sum_{i=1}^{N}\frac{\left(1+s_{D,i}\right)}{2}s_{a,i}-d_{3}{\left(\sum_{i=1}^{N}\frac{\left(1+s_{D,i}\right)}{2}-N_{r}\right)}^{2}-d_{4}N$$

As it has already been mentioned, if a smart grid device is attacked, this might have several negative consequences, such as the disruption of the normal operation of the electricity grid and financial loss. For example, when the attacks target the dynamic energy management (DEM) system [26], they might lead to the under/overestimation of the energy consumption and, thus, monetary loss in energy trading. This is reflected to the first term of the payoff function of the defender. More specifically, $d_{1}$ can be seen as a function of the average cost, $E[C_{i}]$, of under- or overestimating the energy demand of the attacked device, assuming that the later corresponds to an energy consumer. Furthermore, assuming that the DEM management operation is implemented over two stages—the unit-commitment and economic-dispatch stages—the utility generates and reserves the energy supply based on the estimated energy demand of the consumers, while if the energy supply was underestimated, the utility needs to buy the energy difference between the actual and the generated energies in the economic dispatch stage to prevent the undersupply situation [27]. In this case, the cost of under- or overestimating the energy demand of the attacked devices is given by [27,28,29]
(5)$$C_{i}=p_{\mathrm{uc}}\int_{0}^{\delta_{i}}\left(\delta_{i}-r\right)f_{R,i}dr+p_{\mathrm{ed}}\int_{\delta_{i}}^{E_{\max}}\left(r-\delta_{i}\right)f_{R,i}dr,$$
where $f_{R,i}$ is the probability density function of the actual energy consumption, $\delta_{i}$ is the mean energy demand of the *i*-th device, $E_{\max}$ is the maximum energy consumption, and $p_{\mathrm{uc}}$ and $p_{\mathrm{ed}}$ are the energy prices in the unit commitment and economic dispatch stages, respectively. On the other hand, the isolated use of some devices could lead to a nonlinear increase of the energy cost, e.g., when a local energy generator is used [30], which is taken into account by the third term of the defender’s payoff.

Moreover, it is highlighted that the different terms of the players’ payoff do not necessarily correspond to direct monetary loss or gain, but also reflect the potential impact of security risks on the reliable operation of the smart grid, which has an indirect effect on financial loss. Furthermore, it is noted that many of our results could easily be generalized assuming different functions for both $U_{D}$ and $U_{A}$.

## 3. One-Shot Game

In this section, we focus on an one-shot non-cooperative game between the attacker and the defender, which captures one encounter between them. In general, although game theory is based on optimization, it is the appropriate tool when the optimal decision of one entity (player) depends on the decision of the other player. The rules for predicting how a game will be played defines the solution concepts in terms of which the game is understood [31].

### 3.1. Game Formulation

It is assumed that both the attacker and the defender have complete information about each other’s payoff. The attacker attacks θN hosts, where 0≤θ≤1 denotes the portion of the total number of honeypots hosts. It is further assumed that the IPs are dynamically assigned and that all hosts have the same probability to be a honeypot or to be attacked. It is noted that this assumption leads to the optimal performance for the defender, as it has been shown in [32]. In this case, the attacker’s payoff can be directly expressed as a function of $a_{i}$, ϕ, θ, and *N*, i.e.,
(6)$$U_{A}=\tilde{f}\left(a_{1},a_{2},a_{3},\phi ,\theta ,N\right).$$

The attacker aims at maximizing its payoff, thus the corresponding optimization problem can be written as
(7)$$\begin{array}{cc} \max_{\phi }\; & U_{A}\\ s.t.\; & C_{1}:0\leq \phi \leq \phi_{m}, \end{array}$$
where $\phi_{m}$ is the maximum value of ϕ.

Similarly, the payoff of the defender can also be written as a function of ϕ and θ, i.e.,
(8)$$U_{D}=\tilde{g}\left(d_{1},d_{2},d_{3},d_{4},\phi ,\theta ,N\right),$$
while its maximization leads to the following optimization problem.
(9)$$\begin{array}{cc} \min_{\theta ,N}\; & U_{D}\\ s.t.\; & C_{1}:0\leq \theta \leq 1\\ \; & C_{2}:0\leq N\leq N_{\max} \end{array}$$

Based on (Equation 2) and (Equation 4), the payoff of the attacker and the defender can be written as
(10)$$U_{A}=a_{1}\left(1-\theta \right)\phi N-a_{2}\theta \phi N-a_{3}\phi N$$
and
(11)$$U_{D}=d_{1}\theta \phi N-d_{2}\left(1-\theta \right)\phi N-d_{3}{\left(\left(1-\theta \right)N-N_{r}\right)}^{2}-d_{4}N,$$
respectively.

Thus, the game, hereinafter termed as Game 1, that captures this situation consists of the following.

*Game 1*:The set of players S, which includes the attacker and the defender, i.e., S={A,D}The set of actions for each player, i.e, $A_{D}=\left\{\theta \in \left[0,1\right],N\in \left[0,N_{\max}\right]\right\}$ for the defender and $A_{A}=\phi \in \left[0,\phi_{m}\right]$ for the attacker.The payoff functions for each player, i.e., $U_{A}$ and $U_{D}$.

Then, the game can be described by the set $G_{1}$:(12)$$G_{1}:\left\{S,A_{D},A_{A},U_{A},U_{D}\right\}.$$

Based on the definitions of the payoffs and strategies in $G_{1}$, the defender tries to select the total number of hosts and honeypots in order to mitigate the impact of attacks by maximizing its payoff, while the attacker aims at maximizing its payoff by properly selecting the number of hosts that will attack. Moreover, this game can be classified as a sequential one of imperfect information, as the defender first decides the number of hosts (*N*) and the portion of them that corresponds to honeypots (θ), while the attacker has partial knowledge of the defender’s strategy, as the attacker can observe the total number of hosts, but does not know which of them are honeypots [33]. Thus, it is assumed that the two players choose θ and ϕ simultaneously at the beginning of the game, assuming common knowledge about the game (payoffs). As it has already been mentioned, the objective of both players is to maximize their payoffs, which implies that both players are rational [33].

### 3.2. Solution of Game 1

In order to solve the game that has been described in the previous subsection, the concept of Nash Equilibrium (NE) will be used. From the practical point of view, the NE is the optimal decision for a player, e.g., the defender, given that the strategy of the other player, i.e., the attacker, is also optimized. Moreover, if a player decides to optimize their payoff ignoring the payoff of the other player and alleviate from the NE, then they will achieve a worst payoff if the other player sticks to the NE. In conclusion, in the considered framework, a defender’s strategy belongs to the NE if it is is the best reply to the attacker’s strategy, and vice versa. In a NE, “unilateral deviations”, which refer to the case that one player changes its own decision while the others stick to their current choices, do not benefit any of the players [31].

**Definition** **1.**
*The action profile ($\theta^{*}$, $\phi^{*}$, $N^{*}$) is a NE if by deviating from it none of the players can gain anything, i.e.,*
(13)$$\begin{array}{cc} U_{D}\left(\theta^{*},N^{*},\phi^{*}\right) & \geq U_{D}\left(\theta ,N,\phi^{*}\right),\\ U_{A}\left(\theta^{*},N^{*},\phi^{*}\right) & \geq U_{A}\left(\theta^{*},N^{*},\phi \right). \end{array}$$

*Thus, a strategy of each player belongs to the NE if this is a best reply to the strategy of the other player [31].*


To derive the NE, first, the following lemma is provided which reduces the set of the candidate best strategies for the attacker.

**Lemma** **1.**
*In the NE—if it exists—$\phi^{*}\in \left\{0,\phi_{m}\right\}$.*


**Proof.** Let us assume that the set $\left(\theta^{*},N^{*}\neq 0,\phi^{\prime }\right)$ is is a NE and that $\phi^{\prime }\in \left(0,\phi_{m}\right)$. Then, it holds that
(14)$$\left(a_{1}\left(1-\theta \right)N-a_{2}\theta N-a_{3}N\right)\phi^{\prime }\geq \left(a_{1}\left(1-\theta \right)-a_{2}\theta -a_{3}\right)\phi_{m},$$
i.e., $\phi^{\prime }\geq \phi_{m}$, which contradicts the assumption. □

**Theorem** **1.**
*The NE is given by*
(15)$$\begin{array}{cc} (\theta^{*},N^{*},\phi^{*})=\; & \left\{\begin{array}{c} (0,\frac{2d_{3}N_{r}-d_{4}}{2d_{3}},0),\;\mathrm{if}\\ \;0\leq \frac{2d_{3}N_{r}-d_{4}}{2d_{3}}\leq N_{\max}\;\mathrm{and}\;a_{1}\leq a_{3},\\ (0,N_{\max},0),\;\mathrm{if}\\ \;\frac{2d_{3}N_{r}-d_{4}}{2d_{3}}>N_{\max}\;\mathrm{and}\;a_{1}\leq a_{3},\\ (0,0,0),\mathrm{if}\\ \;\frac{2d_{3}N_{r}-d_{4}}{2d_{3}}<0\;\mathrm{and}\;a_{1}\leq a_{3},\\ (\frac{\left(d_{1}+d_{2}\right)\phi_{m}+2d_{3}N_{\max}-2d_{3}N_{r}}{2d_{3}N_{\max}},N_{\max},\phi_{m}),\mathrm{if}\;\\ \;0\leq \frac{\left(d_{1}+d_{2}\right)\phi_{m}+2d_{3}N_{\max}-2d_{3}N_{r}}{2d_{3}}\leq N_{\max}\;\mathrm{and}\;d_{1}\phi_{m}\geq d_{4}\\ \;\mathrm{and}\;\left(a_{1}+a_{2}\right)N_{r}\geq \left(a_{2}+a_{3}\right)N_{\max}+\frac{\left(a_{1}+a_{2}\right)\left(d_{1}+d_{2}\right)}{2d_{3}},\\ (0,N_{r}-\frac{d_{2}\phi_{m}+d_{4}}{2d_{3}},\phi_{m}),\;\mathrm{if}\\ \;d_{1}\phi_{m}<d_{4}\;\mathrm{and}\;a_{1}>a_{3}\;\mathrm{and}\;0<N_{r}-\frac{d_{2}\phi_{m}+d_{4}}{2d_{3}}\leq N_{\max},\\ (0,N_{\max},\phi_{m}),\;\mathrm{if}\\ \;\left(d_{1}+d_{2}\right)\phi_{m}+2d_{3}N_{\max}-2d_{3}N_{r}<0\;\mathrm{and}\\ \;a_{1}>a_{3}\;\mathrm{and}\;N_{r}-\frac{d_{2}\phi_{m}+d_{4}}{2d_{3}}>N_{\max},\\ (0,0,\phi_{m}),\;\mathrm{if}\\ \;\left(d_{1}+d_{2}\right)\phi_{m}+2d_{3}N_{\max}-2d_{3}N_{r}<0\;\mathrm{and}\\ \;a_{1}>a_{3}\;\mathrm{and}\;N_{r}-\frac{d_{2}\phi_{m}+d_{4}}{2d_{3}}<0,\\ ∄,\;\mathrm{elsewhere}. \end{array}\right. \end{array}$$


**Proof.** By using Lemma 1, the values of ϕ that can potentially belong to the NE are ϕ=0 and $\phi =\phi_{m}$. First, let us assume that $\phi^{*}=0$. This can be valid only if
(16)$$\phi_{m}\left(a_{1}\left(1-\theta \right)-a_{2}\theta -a_{3}\right)\leq 0.$$
when $\phi^{*}=0$ then $\theta^{*}=0$. Thus, ϕ=0 can be belong to the equilibrium if $a_{1}\leq a_{3}$. By setting $\frac{\partial U_{D}}{\partial N}=0$, θ=0, and ϕ=0, it holds that
(17)$$N^{*}={\left[\frac{2d_{3}N_{r}-d_{4}}{2d_{3}}\right]}_{0}^{N_{\max}}$$
where ${\left[\cdot \right]}_{0}^{N_{\max}}=\min\left\{\max\left\{\cdot ,0\right\},N_{\max}\right\}$.Next, let us assume that $\phi^{*}=\phi_{m}$. By setting $\frac{\partial U_{D}}{\partial \theta }=0$, it holds that
(18)$$\theta =\frac{\left(d_{1}+d_{2}\right)\phi_{m}+2d_{3}N-2d_{3}N_{r}}{2d_{3}N},$$
while
(19)$$\frac{\partial^{2}U_{D}}{\partial \theta^{2}}=-2d_{3}N^{2}\leq 0,$$
i.e., $U_{D}$ is concave with respect to θ.By assuming that $0<\frac{\left(d_{1}+d_{2}\right)\phi_{m}+2d_{3}N-2d_{3}N_{r}}{2d_{3}N}<1$, it holds that $\frac{\partial U_{D}}{\partial N}\geq 0$ if $d_{1}\phi_{m}\geq d_{4}$. In this case, the value of θ that maximizes $U_{D}$ is given by (Equation 18) and $N=N_{\max}$, as $U_{D}$ is concave with respect to θ and, given the solution of (Equation 18), an increasing function of *N*. Furthermore, $\phi =\phi_{m}$ belongs to the equilibrium if
(20)$$U_{A,\phi =\phi_{m}}\geq U_{A,\phi =0},$$
which can be written as
(21)$$\left(a_{1}+a_{2}\right)N_{r}\geq \left(a_{2}+a_{3}\right)N_{\max}+\frac{\left(a_{1}+a_{2}\right)\left(d_{1}+d_{2}\right)}{2d_{3}}.$$Finally, if $\frac{\left(d_{1}+d_{2}\right)\phi_{m}+2d_{3}N-2d_{3}N_{r}}{2d_{3}N}<0$ (i.e., $U_{D}$ is not maximized for θ>0), from $\frac{\partial U_{D}}{\partial N}=0$ and considering that $\frac{\partial^{2}U_{D}}{\partial N^{2}}\leq 0$, for θ=0 it holds that
(22)$$N^{*}={\left[N_{r}-\frac{d_{2}\phi_{m}+d_{4}}{2d_{3}}\right]}_{0}^{N_{\max}}.$$Apparently, in this case, $\phi =\phi_{m}$ belongs to the equilibrium if $a_{1}>a_{3}$, as then $U_{A,\phi =\max}\geq U_{A,\phi =0}$.Finally, it is noted that θ=1 cannot belong to an equilibrium, as in this case $\phi^{*}=0$. □

**Theorem** **2.**
*The Nash equilibrium of Game 1—if it exists—is unique.*


**Proof.** This can easily be proved by observing that all sets of conditions for each branch of (Equation 15) are mutually exclusive, as can also be verified by the proof of Theorem 1. □

### 3.3. Strategy Selection When NE Does Not Exist

As it can be observed in the previous subsection, the NE does not always exist. Thus, to meet the requirements of practical scenarios, a different framework is required when the NE does not exist. In this case, the strategy of the defender can be chosen by using “maxmin” analysis, which, instead of relying on predictions about choices of other player, it is concerned with maximizing the lowest value the other player can force the player to receive when they know the player’s action [34]. The maxmin value for the defender is defined as
(23)$$\max_{0\leq \theta \leq 1,0\leq N\leq N_{\max}}\min_{0\leq \phi \leq \phi_{m}}\;U_{D}$$

To solve (Equation 23), it needs to be observed that $U_{D}$ is either an increasing or a decreasing value of ϕ, for specific values of θ and *N*. Thus, the attacker can force the defender to receive the lowest value by either choosing $\phi_{m}$ or 0. When $\phi =\phi_{m}$,
(24)$$U_{D}=d_{1}\theta \phi_{m}N-d_{2}\left(1-\theta \right)\phi_{m}N-d_{3}{\left(\left(1-\theta \right)N-N_{r}\right)}^{2}-d_{4}N,$$
while when ϕ=0,
(25)$$U_{D}=-d_{3}{\left(\left(1-\theta \right)N-N_{r}\right)}^{2}-d_{4}N.$$

Thus, (Equation 23) can be rewritten as
(26)$$\begin{array}{cc} \max_{\theta ,N}\; & y\\ s.t.\; & C_{1}:0\leq \theta \leq 1,\\ \; & C_{2}:0\leq N\leq N_{\max},\\ \; & C_{3}:d_{1}\theta \phi_{m}N-d_{2}\left(1-\theta \right)\phi_{m}N-d_{3}{\left(\left(1-\theta \right)N-N_{r}\right)}^{2}-d_{4}N\geq y,\\ \; & C_{4}:-d_{3}{\left(\left(1-\theta \right)N-N_{r}\right)}^{2}-d_{4}N\geq y. \end{array}$$

The aforementioned problem is non-convex and thus difficult to solve this in its current format. To this end, by setting $\theta N=N_{1}$ and $\left(1-\theta \right)N=N_{2}$, it can be written as
(27)$$\begin{array}{cc} \max_{N_{1},N_{2}}\; & y\\ s.t.\; & C_{1}:N_{1}+N_{2}\leq N_{\max},\\ \; & C_{2}:d_{1}\phi_{m}N_{1}-d_{2}\phi_{m}N_{2}-d_{3}{\left(N_{2}-N_{r}\right)}^{2}-d_{4}\left(N_{1}+N_{2}\right)\geq y,\\ \; & C_{3}:-d_{3}{\left(N_{2}-N_{r}\right)}^{2}-d_{4}\left(N_{1}+N_{2}\right)\geq y,\\ \; & C_{4}:N_{1},N_{2}\geq 0. \end{array}$$

The optimization problem in (Equation 27) is a convex one and can be solved by standard convex optimization methods.

## 4. Repeated Game with Uncertainty about the Type of Attacker

In this section, based on the results for the one-shot game, we focus on a more realistic scenario according to which a repeated game is assumed, i.e., the attacker and the defender play the same game more than once.

### 4.1. Game Formulation

Players observe the outcome of the first round before the start of the second round. Payoffs for the entire game are defined as the sum of the payoffs from the previous stages. It is noted that repeated games have a more complex strategic structure than their their one-shot counterparts, as the players’ strategic choices in the following stages are influenced by the outcome of the choices they make in an earlier stage [31].

Furthermore, it is assumed that the defender does not have complete knowledge of the weights of the attacker’s payoff, i.e., $a_{i}$. Among others, this corresponds to the existence of multiple attackers with different preferences or the change of the same attacker’s payoff over time. Then, a multi-stage game which belongs to the class of games known as “multi-stage games with observed actions and incomplete information” is considered. More specifically, it is assumed that there are two types of attackers, namely, *a* and *b*, each of which has different weights. Moreover, the impact of attacks from each type at the defender might be different, which is reflected by the use of wights $d_{i,a}$ and $d_{i,b}$, when the attacks come from the attacker *a* and *b*, respectively. It is assumed that in each time slot, solely all attacks are from the same type of attacker. Similarly to $G_{1}$, the attacker does not have perfect information of the last value of theta selected by the defender, but perfectly knows all former actions. A mixed strategy is assumed for the attacker where the attacker plays according to a probability distribution over the available strategies. Such a randomized behavior can potentially mislead the defender and lead them to reduced performance in terms of achieved average payoff. Thus, to model the interaction between the attacker and the defender, the concept of Bayesian games will be used [31]. In general, in Bayesian games, the term “type” is used to capture the incomplete information. In addition to the actual players in the game, there is a special player called “Nature”. Nature randomly chooses a type for the attacker. It is further assumed that the distribution of Nature’s moves is also unknown.

More specifically, let us assume that the attacker performs an attack to each host with probability $\phi_{i}<\phi_{m}$ with i∈{a,b}. Then, the expected payoff of the attacker is
(28)$$E\left[U_{A_{i}}\right]=a_{1,i}\left(1-\theta \right)\phi_{i}N-a_{2,i}\theta \phi_{i}N-a_{3,i}\phi_{i}N,$$
where E[·] denotes expectation. We assume that $0\leq \phi_{i}\leq \phi_{i,m}$, where $\phi_{i,m}$ is the maximum value of $\phi_{i}$. It is noted that for practical reasons $\phi_{i,m}<1$, as attacking all defender’s hosts would lead to extreme measures from the defender. It is highlighted that hereinafter when ϕ is used will denote the portion of hosts that are attacked (without specifying who is the attacker), while when $\phi_{i}$ is used will denote the probability that the attacker of type *i* attacks each host. Moreover, to avoid redundancy, it is further assumed that $a_{1,a}>a_{3,a}$ and $a_{1,b}>a_{3,b}$, as otherwise if one of the inequalities does not hold true, the attack cannot come from the corresponding type of attacker.

Based on the described attacker’s behavior, the defender’s strategy depends on his belief about the attacker’s type. More specifically, the defender’s belief is defined as a probability distribution over the nodes within his/her information set, conditioned on the fact that this information set has been reached. In other words, it represents how likely this player believes that a certain number of attacks comes from a certain type of the opponent. A system of beliefs is assembled from all individual information sets. For the current game, we only need to define belief for the attacker, i.e., their type. Let 0≤μ≤1 denote the belief of the defender that the attacker is of type *a*.

Considering the above, the expected value for the defender is
(29)$$\begin{array}{cc} \; & E\left[U_{D}\right]=\mu \times \left(d_{1,a}\theta \phi_{a}N-d_{2,a}\left(1-\theta \right)\phi_{a}N-d_{3}{\left(\left(1-\theta \right)N-N_{r}\right)}^{2}-d_{4}N\right)\\ \; & +\left(1-\mu \right)\times (d_{1,b}\theta \phi_{b}N-d_{2,b}\left(1-\theta \right)\phi_{b}N-d_{3}{\left(\left(1-\theta \right)N-N_{r}\right)}^{2}-d_{4}N). \end{array}$$

It is assumed that both players aim at maximizing their expected payoffs. Thus, the game, hereinafter termed as Game 2, that captures this situation consists of the following.

*Game 2*:(i)The set of players S that includes the attacker and the defender, i.e., S={A,D}.(ii)The set of states of nature, denoted by Ω.(iii)The types of the attacker, i.e., the set (a,b).(iv)The set of actions for each player, i.e, $A_{D}=\left\{\theta ,N\right\}$ for the defender and $\left(A_{A_{a}},A_{A_{a}}\right)=\left(\phi_{a},\phi_{b}\right)$ for the attacker of type *a* and *b*, respectively.(v)The expected payoff functions for each player, i.e., $E[U_{A}]$ and $E[U_{A}]$.(vi)The belief μ about the type of the attacker.(vii)The history $h^{t}$ of the game at the *t*-th round.

The game can be described by the set $G_{2}$:(30)$$G_{2}:\left\{S,\Omega ,\left(a,b\right),A_{D},\left(A_{A_{a}},A_{A_{b}}\right),E\left[U_{A}\right],E\left[U_{D}\right],\mu ,h^{t}\right\}$$

### 4.2. Solution of Game 2 Given Updated Beliefs

To solve the situation described in the previous subsection, the Bayesian Nash Equilibrium (BNE) will be used.

**Lemma** **2.**
*In the BNE—if it exists—$\phi_{a}^{*}\in \left\{0,\phi_{a,m}\right\}$ and $\phi_{b}^{*}\in \left\{0,\phi_{b,m}\right\}$.*


**Proof.** The proof is similar to the one of Lemma 1. □

In the following, we analyze BNE based on the assumption that μ is a common prior, i.e., the attacker knows the defender’ belief of μ [35].

**Theorem** **3.**
*The BNE is given by*
(31)$$\begin{array}{cc} \; & (\theta^{*},N^{*},\phi_{a}^{*},\phi_{b}^{*})=\\ \; & \left\{\begin{array}{c} ({\tilde{\theta }}_{1},N_{\max},\phi_{a,m},\phi_{b,m}),\mathrm{if}\\ \;0\leq {\tilde{\theta }}_{1}\leq 1\;\mathrm{and}\;\mu d_{1,a}\phi_{a,m}+d_{1,b}\phi_{b,m}\left(1-\mu \right)\geq d_{4}\\ \;\mathrm{and}\;-\mu \left(d_{1,a}+d_{2,a}\right)\phi_{a,m}+\left(\mu -1\right)\left(d_{1,b}+d_{2,b}\right)\phi_{b,m}+2d_{3}N_{r}\geq \frac{2d_{3}\left(a_{2,a}+a_{3,a}\right)N_{\max}}{a_{1,a}+a_{2,a}}\\ \;\mathrm{and}\;-\mu \left(d_{1,a}+d_{2,a}\right)\phi_{a,m}+\left(\mu -1\right)\left(d_{1,b}+d_{2,b}\right)\phi_{b,m}+2d_{3}N_{r}\geq \frac{2d_{3}\left(a_{2,b}+a_{3,b}\right)N_{\max}}{a_{1,b}+a_{2,b}},\\ (0,N_{r}-\frac{d_{4}+d_{2,b}\phi_{b,m}+d_{2,a}\mu \phi_{a,m}-d_{2,b}\mu \phi_{b,m}}{2d_{3}},\phi_{a,m},\phi_{b,m}),\;\mathrm{if}\\ \;{\tilde{\theta }}_{1}<0\;\mathrm{and}\;0\leq N_{r}-\frac{d_{4}+d_{2,b}\phi_{b,m}+d_{2,a}\mu \phi_{a,m}-d_{2,b}\mu \phi_{b,m}}{2d_{3}}\leq N_{\max},\\ (0,N_{\max},\phi_{a,m},\phi_{b,m}),\;\mathrm{if}\\ \;{\tilde{\theta }}_{1}<0\;\mathrm{and}\;\mathrm{and}N_{r}-\frac{d_{4}+d_{2,b}\phi_{b,m}+d_{2,a}\mu \phi_{a,m}-d_{2,b}\mu \phi_{b,m}}{2d_{3}}>N_{\max},\\ (0,0,\phi_{a,m},\phi_{b,m}),\;\mathrm{if}\\ \;{\tilde{\theta }}_{1}<0\;\mathrm{and}\;\mathrm{and}N_{r}-\frac{d_{4}+d_{2,b}\phi_{b,m}+d_{2,a}\mu \phi_{a,m}-d_{2,b}\mu \phi_{b,m}}{2d_{3}}<0,\\ ({\tilde{\theta }}_{2},N_{\max},\phi_{a,m},0),\;\mathrm{if}\\ \;0\leq {\tilde{\theta }}_{2}\leq 1\;\mathrm{and}\;\mu \phi_{a,m}d_{1,a}\geq d_{4}\;\mathrm{and}\;\frac{\left(a_{1,a}+a_{2,a}\right)\left(2d_{3}N_{r}-\mu \phi_{a,m}\left(d_{1,a}+d_{2,a}\right)\right)}{d_{3}\left(a_{2,a}+a_{3,a}\right)}\geq N_{\max}\\ \;\mathrm{and}\;\frac{\left(a_{1,b}+a_{2,b}\right)\left(2d_{3}N_{r}-\mu \phi_{a,m}\left(d_{1,a}+d_{2,a}\right)\right)}{d_{3}\left(a_{2,a}+a_{3,a}\right)}\leq N_{\max},\\ (0,0,\phi_{a,m},0),\;\mathrm{if}\\ \;{\tilde{\theta }}_{2}<0\;\mathrm{and}\;N_{r}-\frac{d_{4}+\mu d_{2,a}\phi_{a,m}}{2d_{3}}<0,\\ ({\tilde{\theta }}_{3},N_{\max},0,\phi_{b,m}),\mathrm{if}\\ \;0\leq {\tilde{\theta }}_{3}\leq 1\;\mathrm{and}\;\left(\mu -1\right)\phi_{b,m}d_{1,b}\geq d_{4}\;\mathrm{and}\;\frac{\left(a_{1,a}+a_{2,a}\right)\left(2d_{3}N_{r}-\mu \phi_{a,m}\left(d_{1,a}+d_{2,a}\right)\right)}{d_{3}\left(a_{2,a}+a_{3,a}\right)}\leq N_{\max}\\ \;\mathrm{and}\;\frac{\left(a_{1,b}+a_{2,b}\right)\left(2d_{3}N_{r}-\mu \phi_{a,m}\left(d_{1,a}+d_{2,a}\right)\right)}{d_{3}\left(a_{2,a}+a_{3,a}\right)}\geq N_{\max},\\ (0,0,0,\phi_{b,m}),\;\mathrm{if}\\ \;{\tilde{\theta }}_{3}<0\;\mathrm{and}\;N_{r}-\frac{\left(1-\mu \right)\phi_{b,m}d_{2,b}+d_{4}}{2d_{3}}<0,\\ ∄,\;\mathrm{elsewhere}, \end{array}\right. \end{array}$$

*where*
(32)$${\tilde{\theta }}_{1}=\frac{\left(d_{1,a}+d_{2,a}\right)\mu \phi_{a,m}}{2N^{*}d_{3}}+\frac{\left(d_{1,b}+d_{2,b}-\mu d_{1,b}-\mu d_{2,b}\right)\phi_{b,m}+2d_{3}\left(N^{*}-N_{r}\right)}{2N^{*}d_{3}},$$
(33)$${\tilde{\theta }}_{2}=\frac{\mu \phi_{a,m}d_{1,a}+\mu \phi_{a,m}d_{2,a}+2d_{3}N^{*}-2d_{3}N_{r}}{2N^{*}d_{3}},$$
(34)$${\tilde{\theta }}_{3}=\frac{\left(1-\mu \right)\phi_{b,m}d_{1,b}+\left(1-\mu \right)\phi_{b,m}d_{2,b}+2d_{3}N^{*}-2d_{3}N_{r}}{2N^{*}d_{3}},$$
*with the value of $N^{*}$ in (Equation 32)–(Equation 34) being determined by the branch that they appear.*


**Proof.** Three different cases will be considered, namely, $\phi_{a}^{*}=\phi_{a,m}$ and $\phi_{b}^{*}=\phi_{b,m}$, $\phi_{1}^{*}=\phi_{a,m}$ and $\phi_{b}^{*}=0$, and $\phi_{a}^{*}=0$ and $\phi_{b}^{*}=\phi_{b,m}$.First, let us assume that $\phi_{1}^{*}=\phi_{1,m}$ and $\phi_{2}^{*}=\phi_{2,m}$. By setting $\frac{\partial U_{D}}{\partial \theta }=0$, it holds that
(35)$$\theta =\frac{\left(d_{1,a}+d_{2,a}\right)\mu \phi_{a,m}}{2Nd_{3}}+\frac{\left(d_{1,b}+d_{2,b}-\mu d_{1,b}-\mu d_{2,b}\right)\phi_{b,m}+2d_{3}\left(N-N_{r}\right)}{2Nd_{3}}$$
and also it is noted that $\frac{\partial^{2}U_{D}}{\partial \theta^{2}}\leq 0$. By assuming that
(36)$$0<\frac{\left(d_{1,a}+d_{2,a}\right)\mu \phi_{a,m}}{2Nd_{3}}+\frac{\left(d_{1,b}+d_{2,b}-\mu d_{1,b}-\mu d_{2,b}\right)\phi_{b,m}+2d_{3}\left(N-N_{r}\right)}{2Nd_{3}}<1,$$
it is given that $\frac{\partial U_{D}}{\partial N}\geq 0$ if $\mu d_{1,a}\phi_{a,m}+d_{1,b}\phi_{b,m}\left(1-\mu \right)\geq d_{4}$. In this case, the value of θ that maximizes $U_{D}$ is given by (Equation 36) and $N=N_{\max}$. Moreover, $\phi_{a}=\phi_{a,m}$ and $\phi_{b}=\phi_{b,m}$ belong to the equilibrium if
(37)$$U_{A_{a},\phi_{a}=\phi_{a,m}}\geq U_{A,\phi_{a}=0}$$
and
(38)$$U_{A_{b},\phi_{b}=\phi_{b,m}}\geq U_{A,\phi_{b}=0},$$
respectively, which can be written as
(39)$$-\mu \left(d_{1,a}+d_{2,a}\right)\phi_{a,m}+\left(\mu -1\right)\left(d_{1,b}+d_{2,b}\right)\phi_{b,m}+2d_{3}N_{r}\geq \frac{2d_{3}\left(a_{2,a}+a_{3,a}\right)N_{\max}}{a_{1,a}+a_{2,a}}$$
and
(40)$$-\mu \left(d_{1,a}+d_{2,a}\right)\phi_{a,m}+\left(\mu -1\right)\left(d_{1,b}+d_{2,b}\right)\phi_{b,m}+2d_{3}N_{r}\geq \frac{2d_{3}\left(a_{2,b}+a_{3,b}\right)N_{\max}}{a_{1,b}+a_{2,b}},$$
respectively.On the other hand, assuming that
(41)$$\frac{\left(d_{1,a}+d_{2,a}\right)\mu \phi_{a,m}}{2N_{\max}d_{3}}+\frac{\left(d_{1,b}+d_{2,b}-\mu d_{1,b}-\mu d_{2,b}\right)\phi_{b,m}+2d_{3}\left(N-N_{r}\right)}{2Nd_{3}}<0,$$
from $\frac{\partial U_{D}}{\partial N}=0$ it holds that
(42)$$N=N_{r}-\frac{d_{4}+d_{2,b}\phi_{b,m}+d_{2,a}\mu \phi_{a,m}-d_{2,b}\mu \phi_{b,m}}{2d_{3}}.$$Apparently, $\phi_{a}^{*}=\phi_{a,m}$ and $\phi_{b}^{*}=\phi_{b,m}$ belong to the equilibrium if $a_{1,a}>a_{3,a}$ and $a_{1,b}>a_{3,b}$, as then $U_{A_{a},\phi_{a}=\phi_{a,m}}\geq U_{A_{a},\phi_{a}=0}$ and $U_{A_{b},\phi_{b}=\phi_{b,m}}\geq U_{A_{b},\phi_{a}=0}$, respectively.Next, it is assumed that $\phi_{a}^{*}=\phi_{a,m}$ and $\phi_{b}^{*}=0$. By setting $\frac{\partial U_{D}}{\partial \theta }=0$ and making similar observations for the second derivative of $U_{D}$ with respect to θ as with the previous case, it holds that
(43)$$\theta =\frac{\mu \phi_{a,m}d_{1,a}+\mu \phi_{a,m}d_{2,a}+2d_{3}N-2d_{3}N_{r}}{2Nd_{3}}.$$By assuming that
(44)$$0<\frac{\mu \phi_{a,m}d_{1,a}+\mu \phi_{a,m}d_{2,a}+2d_{3}N-2d_{3}N_{r}}{2Nd_{3}}<1,$$
it can be shown that $\frac{\partial U_{D}}{\partial N}\geq 0$ if $\mu \phi_{a,m}d_{1,a}\geq d_{4}$. In this case, the value of θ that maximizes $U_{D}$ is given by (Equation 43) and $N=N_{\max}$. Moreover, $\phi_{a}=\phi_{a,m}$ and $\phi_{b}=0$ belong to the equilibrium if
(45)$$U_{A_{a},\phi_{a}=\phi_{a,m}}\geq U_{A,\phi_{a}=0},$$
and
(46)$$U_{A_{b},\phi_{b}=\phi_{b,m}}\leq U_{A,\phi_{b}=0},$$
respectively, which can be written as
(47)$$\frac{\left(a_{1,a}+a_{2,a}\right)\left(2d_{3}N_{r}-\mu \phi_{a,m}\left(d_{1,a}+d_{2,a}\right)\right)}{d_{3}\left(a_{2,a}+a_{3,a}\right)}\geq N_{\max}$$
and
(48)$$\frac{\left(a_{1,b}+a_{2,b}\right)\left(2d_{3}N_{r}-\mu \phi_{a,m}\left(d_{1,a}+d_{2,a}\right)\right)}{d_{3}\left(a_{2,a}+a_{3,a}\right)}\leq N_{\max},$$
respectively. If
(49)$$\frac{\mu \phi_{a,m}d_{1,a}+\mu \phi_{a,m}d_{2,a}+2d_{3}N-2d_{3}N_{r}}{2Nd_{3}}<0,$$
from $\frac{\partial U_{D}}{\partial N}$ it holds that
(50)$$N={\left[N_{r}-\frac{d_{4}+\mu d_{2,a}\phi_{a,m}}{2d_{3}}\right]}_{0}^{N_{\max}}$$Apparently, if
(51)$$N_{r}-\frac{d_{4}+\mu d_{2,a}\phi_{a,m}}{2d_{3}}>0,$$
$\phi_{b}^{*}=0$ cannot belong to the equilibrium, as having assumed that $a_{1,b}>a_{3,b}$, it leads to $U_{A_{b},\phi_{b}=\phi_{b,m}}>U_{A_{b},\phi_{a}=0}$.Finally, assuming that $\phi_{a}^{*}=0$ and $\phi_{b}^{*}=\phi_{b,m}$, similar steps can be followed to find the equilibrium, which result in (Equation 43), (Equation 47), and (Equation 48) being replaced by
(52)$$\theta =\frac{\left(1-\mu \right)\phi_{b,m}d_{1,b}+\left(1-\mu \right)\phi_{b,m}d_{2,b}+2d_{3}N^{*}-2d_{3}N_{r}}{N^{*}2d_{3}}$$
(53)$$\frac{\left(a_{1,a}+a_{2,a}\right)\left(2d_{3}N_{r}-\mu \phi_{a,m}\left(d_{1,a}+d_{2,a}\right)\right)}{d_{3}\left(a_{2,a}+a_{3,a}\right)}\leq N_{\max}$$
and
(54)$$\frac{\left(a_{1,b}+a_{2,b}\right)\left(2d_{3}N_{r}-\mu \phi_{a,m}\left(d_{1,a}+d_{2,a}\right)\right)}{d_{3}\left(a_{2,a}+a_{3,a}\right)}\geq N_{\max},$$ □respectively.

**Theorem** **4.**
*The BNE of Game 2—if it exists—is unique.*


**Proof.** This can easily be proved by observing that all sets of conditions for each branch are mutually exclusive, as it can also be verified by the proof of Theorem 3. □

### 4.3. Update of Belief

As the game has only two players and only the defender needs to maintain its belief at any point in time, the defender’s belief at stage *t* is defined as [33,35]
(55)$$\mu^{t}=P\left(A_{a}|h^{t}\right)$$
and
(56)$$1-\mu^{t}=P\left(A_{b}|h^{t}\right),$$
where $P(A_{i}|\phi )$ is the probability that when the portion of attacked hosts is ϕ, the type of attacker is *i*. Moreover, $h^{t}$ is the history profile of the attacker, defined as a vector that contains the actions of the attacker, i.e.,
(57)$$h^{t}=\left(\phi^{1},...,\phi^{t-1}\right).$$

The belief can be determined by using the Bayes’ rule, i.e.,
(58)$$\mu^{t+1}=\frac{P\left(\phi^{t}|A_{a},h^{t}\right)P\left(A_{a}\right)}{P(\phi^{t}|h^{t})},$$
which can be written as
(59)$$\mu^{t+1}=\frac{P\left(\phi^{t}|A_{a},h^{t}\right)P\left(A_{a}\right)}{P\left(\phi^{t}|A_{a},h^{t}\right)P\left(A_{a}\right)+P\left(\phi^{t}|A_{b},h^{t}\right)P\left(A_{b}\right)}.$$

Observing new actions $\phi^{t}$, the posterior belief $\mu^{t+1}$ via Bayesian updates can be estimated as
(60)$$\mu^{t+1}=\frac{\mu^{t}P\left(\phi^{t}|A_{a},h^{t}\right)}{\mu^{t}P\left(\phi^{t}|A_{a},h^{t}\right)+\left(1-\mu^{t}\right)P\left(\phi^{t}|A_{b},h^{t}\right)}.$$

It is further assumed that each player believes that their opponent is playing according to the BNE. Thus, $P(\phi^{t}|A_{a},h^{t})$ and $P(\phi^{t}|A_{b},h^{t})$ can be calculated using the binomial distribution formula by
(61)P(ϕt|Aa,ht)=NϕtN(ϕat*)ϕtN(1−ϕat*)N(1−ϕt)
and
(62)P(ϕt|Ab,ht)=NϕtN(ϕbt*)ϕtN(1−ϕbt*)(1−ϕt)N,
where nk=n!k!(n−k)!. Considering the above, and as the term NϕtN appears in both $P(\phi |A_{a},h^{t})$ and $P(\phi^{t}|A_{b},h^{t})$, (Equation 60) can be rewritten as
(63)$$\mu^{t+1}=\frac{\mu^{t}{\left(\phi_{a}^{t*}\right)}^{\phi^{t}N}{\left(1-\phi_{a}^{t*}\right)}^{N(1-\phi^{t})}}{\mu^{t}{\left(\phi_{a}^{t*}\right)}^{\phi^{t}N}{\left(1-\phi_{a}^{t*}\right)}^{N(1-\phi^{t})}+\left(1-\mu^{t}\right){\left(\phi_{b}^{t*}\right)}^{\phi^{t}N}{\left(1-\phi_{b}^{t*}\right)}^{(1-\phi^{t})N}}.$$

## 5. Simulation Results & Discussion

To study the behavior of our model, a simulation environment was implemented in Python. Three experiments have been carried out to study the player’s strategies and the overall system behavior for the one-shot game and for the repeated game as well as in the case that NE does not exist.

### 5.1. One-Shot Game

The parameters that were used for the one-shot game are provided in Table 2. It should be noted that the simulation results do not depend on the exact values of the weights ($a_{i}$ and $d_{i}$) but on the ratio among them; thus, the utilized values of weights are normalized to a common value. In this experiment, we compare the optimal strategy for the attacker and the defender with 2000 random solutions in order to verify that the equilibrium indeed yields the maximum payoff, considering that the opponent always chooses the best strategy.

The provided Figure 2 and Figure 3 verify that the payoffs of both players are optimal when the game reaches its equilibrium state. The red bullet in each graph points to the payoff in the equilibrium state. In more detail, Figure 2 shows that the payoff achieved in the equilibrium state (red bullet) is higher compared to 2000 random strategies ϕ, assuming that Nθ remains at the optimal state. Similarly, the payoff achieved for the defender in Figure 3 is higher compared to $2000N_{\max}$ random combinations of Nθ, assuming that the opponent always chooses the best possible strategy. Moreover, it is notable that the payoffs follow a specific pattern when *N* remains constant and θ varies.

### 5.2. Max-Min Solution in the One-Shot Game

The second experiment examines the situation in which the game parameters do not result in an equilibrium, thus the defender applies a max-min analysis to maximize the worst-case scenario as described in Equation (Equation 27). The parameters of this experiment are provided in Table 3. The convex optimization problem of Equation (Equation 27) was solved by employing the CVXPY Python library [36,37].

Figure 4 depicts the maximum worst-case payoff that corresponds to the solution received by Equation (Equation 27). This solution is compared to the worst-case payoffs that are received for different values of Nθ. The results prove that the defender successfully chooses the best possible strategy that yields the maximum payoff, assuming that the attacker always chooses the best strategy. Similar trends for various values of *N* are noticed, as with the first experiment.

### 5.3. Repeated Game

Finally, a third experiment was carried out in order to evaluate and study the repeated game. The main characteristic of this game is that the defender has imperfect information about the attacker’s strategy, i.e., the identity of the actual kind of attacker that is hidden behind each attack. The simulation parameters are provided in Table 4.

The experiment realizes two type of defenders that correspond to deployment strategies and preferences of defenders deploying production or research honeypots. In our example, type *a* corresponds to a defender that deploys production honeypots and type *b* corresponds to a defender that deploys research honeypots. This is justified as the defender that deploys production honeypots cares more about the impact on the production equipment, meaning that the damage that an attack would cause against real devices would be greater than the benefit that the defender would enjoy if this attack would be against a honeypot, i.e., $d_{a,1}>d_{a,2}$. On the contrary, a defender that deploys research honeypots cares more about attracting attackers, meaning that the benefit for each attack against honeypots would be greater than the damage against a real device would cause, i.e., $d_{b,1}<d_{b,2}$.

As with the one-shot experiment, Figure 5 and Figure 6 illustrate the achieved payoff of player’s equilibrium, in respect to 2000 random solutions for the attacker and $2000N_{\max}$ random solutions for the defender. Once again, the red bullet on each of these graphs depicts the equilibrium state. It is validated from Figure 5 that the attacker payoff drops if the attacker deviates from the optimal solution derived from the equilibrium. The same behavior is also noticed in Figure 6 for the defender.

Finally, Figure 7 and Figure 8 study the defender’s belief about the actual type of the attacker. In particular, Figure 7 compares the belief of the defender about the attacker’s type with the actual type of the attacker. It is assumed that 1 represents type *a* and 0 represents type *b*. It is shown that the defender successfully identifies the attacker’s type in round 4 and does not change their belief, as the attacker’s behavior remains the same throughout the game

Figure 8 depicts the players’ payoff through time. It is evident that the defender’s payoff increases as they approach the actual type of the attacker. The defender’s payoff exceeds the payoff of its opponent and gets maximized after round 4, when the defender is confident enough about the real type of the attacker.

## 6. Conclusions

In this paper, the efficient use of honeypots has been considered with the aim of mitigating the impact of attacks to smart grid infrastructure. More specifically, the interaction of an attacker and a defender has been investigated, who both aim at maximizing their payoffs by optimizing the deployment of attacks and honeypots, respectively. Two different games have been considered, namely, a one-shot one with perfect knowledge of the players’ payoff and a repeated one with uncertainty about the payoff of the attacker. The Nash Equilibrium and the Bayesian Nash Equilibrium have been derived for the first and the second game, respectively, as well as the corresponding conditions, while the Equilibria uniqueness has been proved. Moreover, an alternative framework has also been provided for the case that an Equilibrium does not exist, which can be seen as the optimization of the worst-case scenario, as it is based on the maximization of the lowest value the attacker can force the defender to receive when they know the defender’s action. Simulation results validated the analytical results of the equilibrium for both the attacker and the defender, for both games. Furthermore, the derived solution for case that the equilibrium does not exist has also also been evaluated. Finally, concerning the repeated game, it has been shown that the defender successfully identifies the attacker’s type, thus maximizing its payoff throughout the game.

The proposed theoretical framework in the considered analysis facilitates the investigation of the potential benefits of using honeypots to enhance security in smart grids and creates opportunities for future research on this topic. For example, the use of more complicated payoffs can be explored, taking into account the particularities of different case studies. Moreover, further research is also needed in order to specify the long-term monetary gain of capturing attacks of a certain type by the utilized honeypots. Finally, the results can be extended to the case of more than two attackers types, while also considering uncertainty for the type of the defender.

## Figures and Tables

**Figure 1 sensors-20-04199-f001:**
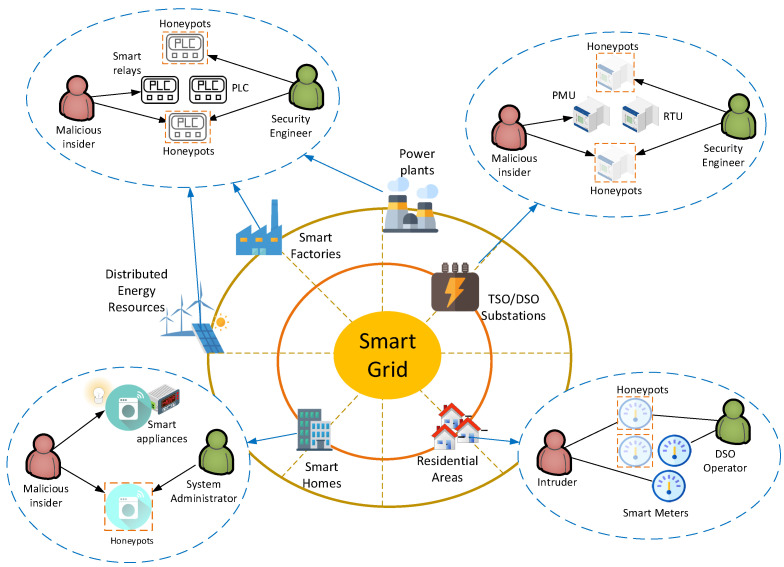
Depiction of various threats and possible honeypot deployments in smart grids.

**Figure 2 sensors-20-04199-f002:**
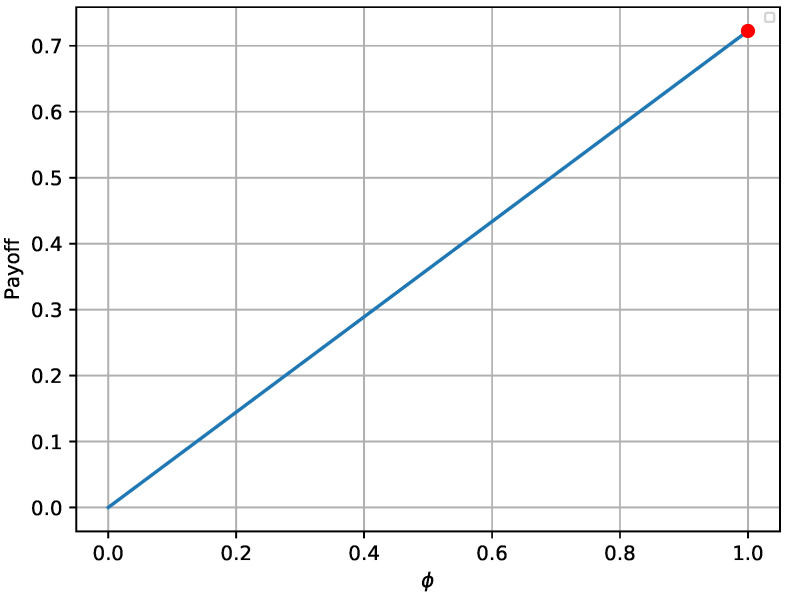
Attacker’s payoff for different strategies in the one-shot game.

**Figure 3 sensors-20-04199-f003:**
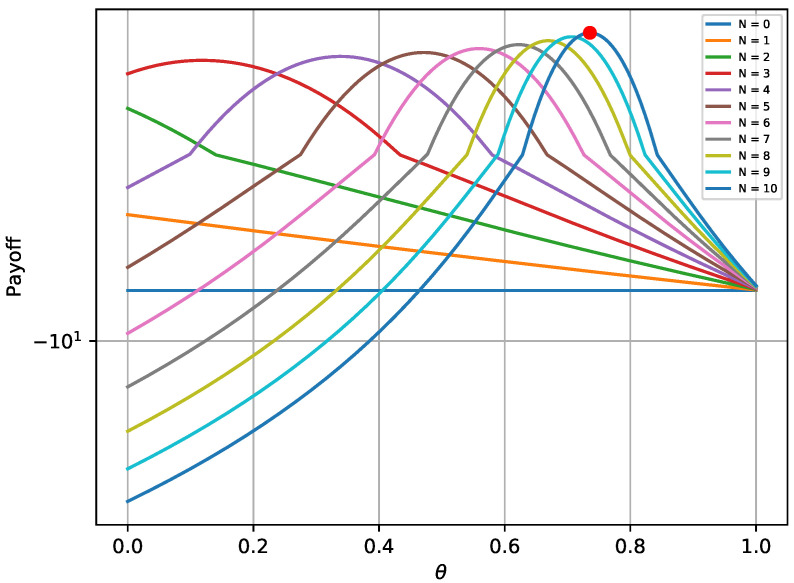
Defender’s payoff for different strategies in the one-shot game.

**Figure 4 sensors-20-04199-f004:**
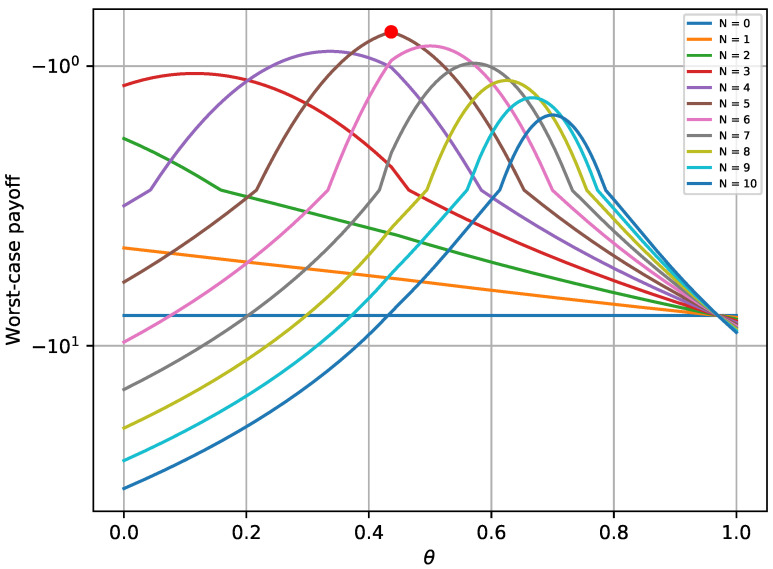
Defender’s worst-case payoff when equilibrium does not exist.

**Figure 5 sensors-20-04199-f005:**
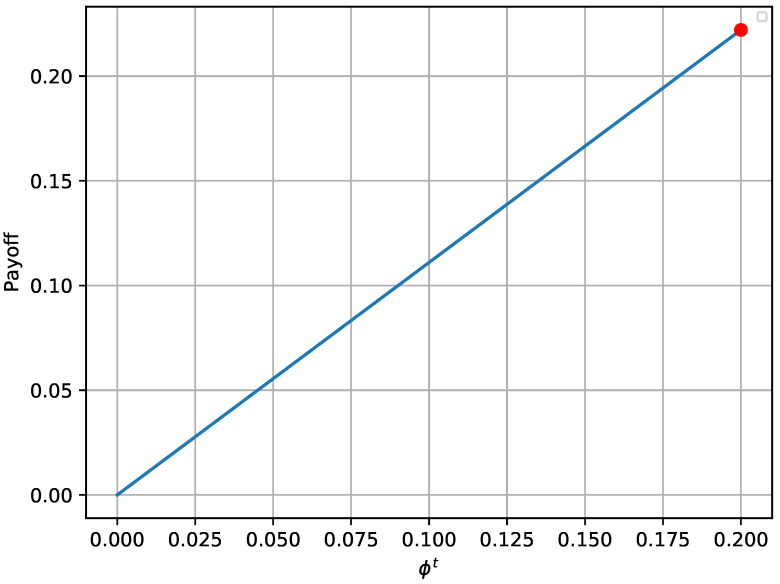
Attacker’s payoff for different strategies in a single turn of the repeated game.

**Figure 6 sensors-20-04199-f006:**
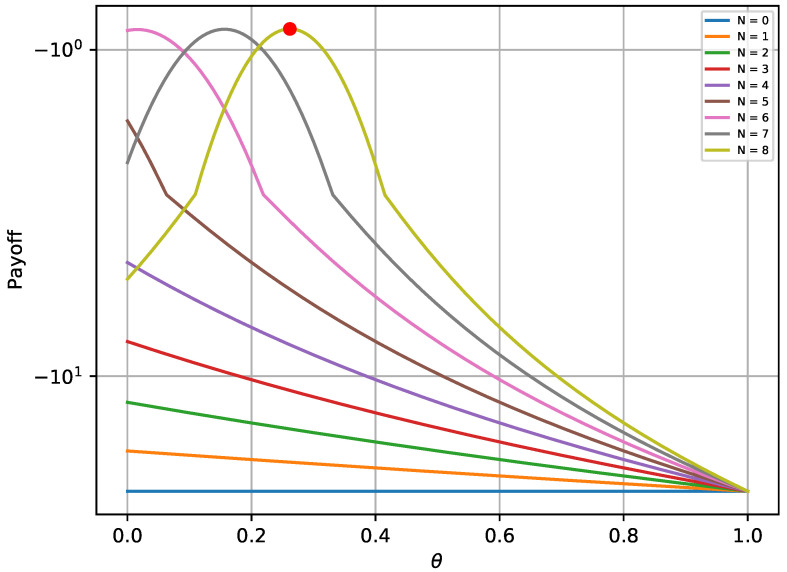
Players’ payoff for different strategies in a single turn of the repeated game.

**Figure 7 sensors-20-04199-f007:**
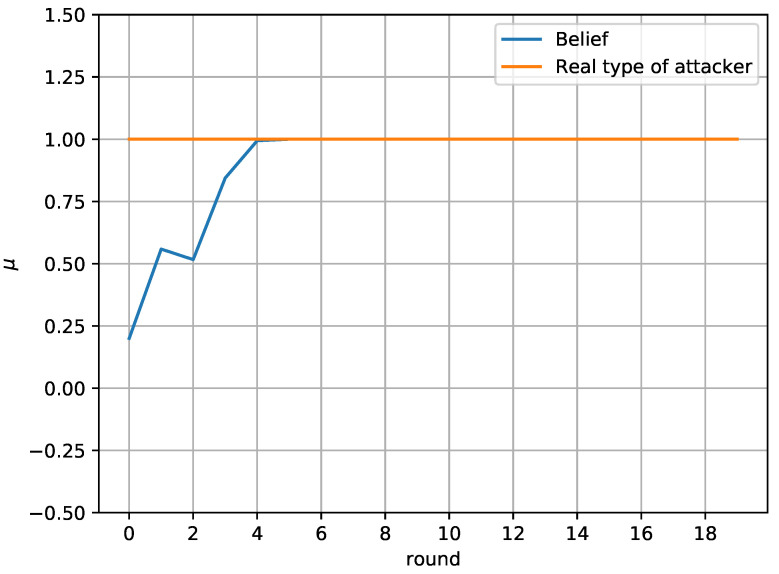
Defender’s belief in the repeated game.

**Figure 8 sensors-20-04199-f008:**
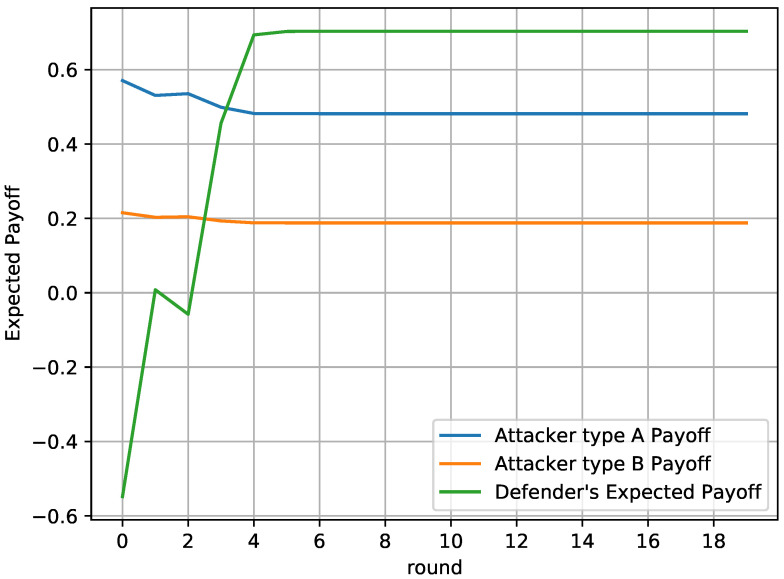
Defender’s payoff in the repeated game.

**Table 1 sensors-20-04199-t001:** Notation.

Parameter	Definition
*A*	attacker
*D*	defender
$s_{A,i}$	strategy of the attacker for the *i*-th host
$s_{D,i}$	strategy of the defender for the *i*-th host
$N_{r}$	number of real devices
$N_{\max}$	total number of available hosts
*N*	sum of connected real devices and honeypots
$a_{i}$	different terms’ weights of attacker’s payoffs
$d_{i}$	different terms’ weights of defender’s payoffs
θ	portion of the number of hosts (*N*) that are honeypots
ϕ	portion of the number of hosts (*N*) that are attacked
$\phi_{m}$	the maximum portion of the number of hosts (*N*) that are attacked
$U_{i}$	payoff of player *i*
f(·),g(·),$\tilde{f}\left(\cdot \right)$,$\tilde{g}\left(\cdot \right)$	functions of (·)
S	set of players
$A_{i}$	set of actions for player *i*
*y*, $N_{1}$, $N_{2}$	auxiliary variables
E[·]	expected value of [·]
P[·]	probability of the event [·]
*a*, *b*	the two types of attacker
$A_{j}$	attacker of type *j*
$a_{i,j}$	weight’s of attacker’s payoff when he is of type j∈{a,b}
$d_{j}$	weight’s of attacker’s payoff when he is of type j∈{a,b}
μ	belief that the attacker is of type *a*
$\phi_{i}$	probability of attacking each host for the attacker of type *i*.
$\phi_{i,m}$	maximum value of the probability of attacking each host for the attacker of type *i*
Ω	states of the nature
*t*	round of the game in a repeated game
$G_{i}$	game *i*
$h^{t}$	history of the game after *t*-th play
${\left(\cdot \right)}^{*}$	(·) belongs to the NE
$C_{i}$	cost of under or over estimating the demand of the *i*-th device
$f_{R,i}$	the probability density function of the actual energy consumption
$\delta_{i}$	the mean energy demand of the *i*-th device
$E_{\max}$	the maximum energy consumption
$p_{\mathrm{uc}}$	energy price in the unit commitment stage
$p_{\mathrm{ed}}$	energy price in the economic-dispatch stage

**Table 2 sensors-20-04199-t002:** Simulation parameters for the one-shot game.

Parameter	Value
$N_{r}$	3
$N_{\max}$	10
$\phi_{\max}$	1
$a_{\left\{1,2,3\right\}}$	[0.76,0.01,0.10]
$d_{\left\{1,2,3,4\right\}}$	[0.03,0.40,0.45,0.01]
Random solutions for θ	2000
Random solutions for ϕ	2000

**Table 3 sensors-20-04199-t003:** Simulation parameters for the one-shot game when equilibrium does not exist.

Parameter	Value
$N_{r}$	3
$N_{\max}$	10
$\phi_{\max}$	1
$a_{\left\{1,2,3\right\}}$	[0.81,0.01,0.06]
$d_{\left\{1,2,3,4\right\}}$	[0.31,0.24,0.81,0.14]
Random solutions for θ	2000

**Table 4 sensors-20-04199-t004:** Simulation parameters for the repeated game.

Parameter	Value
Number of rounds	50
$N_{r}$	6
$N_{\max}$	8
$\phi_{a,\max}$	0.6
$\phi_{b,\max}$	0.2
$a_{a\{1,2,3\}}$	[0.48,0.46,0.10]
$a_{b\{1,2,3\}}$	[0.39,0.48,0.02]
$d_{a\{1,2\}}$	[0.70,0.04]
$d_{b\{1,2\}}$	[0.04,0.68]
$d_{3},d_{4}$	0.77,0.006

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
