# Peer review of "Game Theoretic Honeypot Deployment in Smart Grid"

_sensors, 2020, doi:10.3390/s20154199_

Round 1
Reviewer 1 Report
The authors have proposed a game theoretic honeypot deployment strategy to maximize both attackers' and defenders' payoff. The idea is very interesting and the analysis is quite rigorous. However, I have the following comments on this work:
- It will be good if the authors can demonstrate how the strategies are impacting smart grid applications. For example, how much financial loss will happen in energy trading or electricity billing. The connection to smart grid does not seem very strong, this can be further improved.
- Can these strategies be applied to false data injection attack scenarios, for example, as in "Impact of optimal false data injection attacks on local energy trading in a residential microgrid", ICT Express.
- Can the authors compare the security performance with other research papers on honeypot deployment?
- What is the physical significance of Nash Equilibrium in this scenario? The authors could have maximized payoffs for either attacker or defender, why both at the same time?
Author Response
Please see the attachement.

Reviewer 2 Report
The paper studies attacks in smart grids using honeypots. The main contribution is the use of game theory for mitigating attacks in smart grids. The paper is well-written and well-organized. The contributions are significant and the discussion is lucid. Few minor points to improve the presentation of the paper:
1) The paper is heavily mathematical. Some background on the theory would be useful for practitioners like me.
2) A separate related work section to include some of the key works in the field. Perhaps include a table to show how your work compares to prior research.
Reviewer 3 Report
The paper proposed a game-theoretic approach for honeypot deployment in the smart grid for addressing the cyber-physical issues. The paper is well written and the results are promising.
To improve the manuscript the following issues are suggested:
- The attack model can be added in details in the system model.
- Future work can be added to the con conclusion.
- The references can be added: a) Wei, Longfei, Arif I. Sarwat, Walid Saad, and Saroj Biswas. "Stochastic games for power grid protection against coordinated cyber-physical attacks." IEEE Transactions on Smart Grid 9, no. 2 (2016): 684-694. b) Tian, Wen, Xiaopeng Ji, Weiwei Liu, Guangjie Liu, Jiangtao Zhai, Yuewei Dai, and Shuhua Huang. "Prospect Theoretic Study of Honeypot Defense Against Advanced Persistent Threats in Power Grid." IEEE Access 8 (2020): 64075-64085.
Round 2
Reviewer 1 Report
The authors have addressed all my comments adequately. Please make sure to proof-read the paper and correct typos. For example, 'innovative' in line 17 and treats -->'threats' in line 33 of the introduction.